

# Loss of nitrogen via anaerobic ammonium oxidation (anammox) in the California current system during the Quaternary

Zoë R. van Kemenade [1], Zeynep Erdem [1], Ellen C. Hopmans [1], Jaap S. Sinninge Damsté [1,2], Darci Rush [1]

[1] NIOZ Royal Netherlands Institute for Sea Research, PO Box 59, 1790 AB, Den Burg, The Netherlands
[2] Department of Earth Sciences, Utrecht University, Princetonlaan 8a, 3584 CB, Utrecht, the Netherlands

*Correspondence to*: Zoë R. van Kemenade (zoe.van.kemenade@nioz.nl)

**Abstract.** The California current system (CCS) hosts one of the largest oxygen minimum zones (OMZs) in the world: the Eastern North Pacific (ENP) OMZ, which is dissociated into a subtropical and tropical region (i.e., the ESTNP and ETNP). In the modern ENP OMZ, bioavailable nitrogen (N) is lost via denitrification and anaerobic ammonium oxidation (anammox). Even so, paleo-reconstructions of N-loss have focused solely on denitrification. Fluctuations in bulk sedimentary $\delta^{15}N$ over glacial-interglacial cycles have been interpreted to reflect variations in denitrification rates in response to ETNP OMZ intensity changes. This $\delta^{15}N$ signal is thought to be transported northwards to the ESTNP OMZ. Here, we present the first CCS sedimentary record of ladderane lipids, biomarkers for anammox, located within the ESTNP OMZ (32°N; 118°W). Over the last two glacial terminations (~160 cal ka BP), ladderane concentrations were analysed in combination with the index of ladderanes with five cyclobutane moieties (NL$_5$), short-chain (SC) ladderane degradation products, and productivity proxies. This shows that: 1) ladderanes derived from anammox bacteria living within the ESTNP OMZ water column; 2) ladderanes were continuously present, with relatively high concentrations during both glacial- and interglacial-periods, showcasing the ESTNP OMZ must have retained an anoxic core in which N-loss occurred; and 3) anammox abundance appears to have been driven both by OM-remineralization and advection changes, which regulated nutrient and oxygen levels. Our study shows that anammox was an important feature in the CCS and provides a more holistic picture of N-loss dynamics and the development of the ESTNP OMZ over glacial-interglacial cycles. Lastly, ladderanes were also detected in 160–500 cal ka BP sediments (15.7–37.5 mbsf; analysed at a low temporal resolution), highlighting their potential as anammox biomarkers in relatively deeper buried sediments for future studies.

## 1 Introduction

The California current system (CCS) is one of four major Eastern Boundary upwelling systems (EBUS). In EBUS, wind-driven offshore advection of surface waters causes deeper, cold, nutrient-rich waters to be upwelled into the photic zone, fuelling primary productivity (e.g., Bakun and Nelson, 1991). Consequently, the CCS is one of the world's most productive oceanic regions, with year-round upwelling, resulting in high primary production rates (Huyer, 1983; Dorman and Winanat, 1995). In the CCS, the respiration of sinking organic matter (OM), in combination with limited ventilation of the North Pacific intermediate waters (Reid and Mantyla, 1978; Sonnerup et al., 1999; Fine et al., 2001), results in the formation of the



Eastern North Pacific oxygen minimum zone (ENP OMZ). The ENP is divided into the Eastern tropical North Pacific
(ETNP; 0–25°N; 75–180°W) and Eastern subtropical North Pacific (ESTNP; 25–52°N; 75–180°W) OMZs.
The suboxic/anoxic conditions of OMZs cause the marine nitrogen (N) cycle to shift towards two processes that
result in the loss of bioavailable N through the production of dinitrogen gas ($N_2$): 1) anaerobic ammonium oxidation
(anammox) and 2) denitrification. Anammox is the oxidation of ammonium ($NH_4^+$) to $N_2$ using $NO_2^-$ as the terminal electron
acceptor (van de Graaf et al., 1997, 1995), and is performed in the marine water column by anammox bacteria of the genus
'*Ca.* Scalindua' (Kuypers et al., 2003). Anammox bacteria are chemolithoautotrophs and use carbon dioxide ($CO_2$) as their
carbon source. Denitrification is the stepwise reduction of nitrate ($NO_3^-$), to nitrite ($NO_2^-$), to $N_2$ (Kuenen and Robertson,
1987) and is performed by a wide range of organisms, most of which are heterotrophs. During denitrification, nitrous oxide
($N_2O$) can be released as an intermediate product (Kuenen and Robertson, 1987), which has a global warming potential 265
times that of $CO_2$ (Vallero, 2019).
While permanent OMZs contribute to only 8 % of the total oceanic area (Paulmier and Ruiz-Pino, 2009), they are
responsible for 20–50 % of total global N loss (Gruber, 2004; Codispoti et al., 2001). Decreased N availability in OMZs may
limit primary producers, and hence, the uptake of $CO_2$ into the organic matter (OM) pool. This may reduce the efficiency of
the ocean's biological pump, which exports organic C from the euphotic zone to the sea floor. Thus, OMZs not only have a
disproportionately large impact on the marine nitrogen cycle, but changes in N-loss dynamics may also feed back into the
carbon cycle.
The ENP OMZ is expanding both vertically (shoaling towards the ocean's surface; Bograd et al., 2008) and
horizontally (Zhou et al., 2022) with present-day climate change. This follows observed trends of overall deoxygenation of
the North Pacific since the 1960's (Whitney et al., 2007; Stramma et al., 2010; Pierce et al., 2012; Smith et al., 2022), linked
to anthropogenically-induced ocean warming as a response to increased greenhouse gas emissions (Laffoley and Baxter,
2019). As a result of the decreasing dissolved oxygen (DO) concentrations, denitrification has been shown to increase in the
North Pacific over the last decades (Peters et al., 2018; White et al., 2019). Vertical expansion and intensification of the ENP
OMZ have also occurred in the absence of anthropogenic influences in the past, as recorded by redox-sensitive trace metals
in the sedimentary archive (Wang et al., 2020). This is thought to be caused by changes in DO concentrations during glacial-
interglacial transitions (terminations). Model simulations indicate that during glacials, cooling of the polar regions led to a
more restrained and intensified Hadley cell (Nicholson and Flohn, 1981). This is thought to have caused southward transport
of high-oxygen, nutrient-rich North Pacific Intermediate Water (NPIW; Herguera et al., 2010) and limited northward
advection of the warm, oxygen-poor California undercurrent (CU; Fig. 1), resulting in a more oxygenated OMZ. During
interglacials, the oxygen deficiency in the OMZ is thought to have increased due to enhanced advection of the warm,
oxygen-depleted waters of the CU originating from the tropics ((Lembke-Jene et al., 2018; Hendy and Kennett, 2003), water





column stratification (Wang et al., 2020), and enhanced upwelling of nutrient-rich waters (Choumiline et al., 2019). These
previous glacial-interglacial transitions may be considered as analogues for the effect of future climate change on the N-
cycle.

65       In the CCS, enriched isotope ratio values of bulk sedimentary nitrogen ($\delta^{15}N$) during interglacial periods have been

interpreted to reflect increased denitrification in response to OMZ intensification (e.g., Kienast et al., 2002; Kemp et al.,
2003; Liu et al., 2005). Sedimentary $\delta^{15}N$ values are governed by the isotopic fractionation ($\varepsilon$) induced by biological
transformations and can be used to infer past N-cycling. For water column denitrification, the production of $N_2$ induces an
isotope fractionation effect of +20 to +30 ‰ on the residual nitrogen (Ryabenko, 2013; Sigman and Fripiat, 2019).
Enrichment cultures of anammox have, however, recently shown that *Ca.* Scalindua spp. also induces an isotope
fractionation effect of +16 to +30 ‰ (Kobayashi et al., 2019). Although anammox occurs in the modern North Pacific
oxygen deficient waters (Rush et al., 2012a; Peng et al., 2015; Sollai et al., 2015; Hamasaki et al., 2018), and anammox is
reported to be the dominant N-loss process in the Eastern Tropical South Pacific (ESTP; Galán et al., 2009; Thamdrup et al.,
2006; Hamersley et al., 2007), to the best of our knowledge, there are no reconstructions on the occurrence of anammox in
the sediment archive of the CCS. Moreover, a long-standing conundrum is the discrepancy between the timing of enriched
$\delta^{15}N$ values, and enhanced marine productivity, especially north of the ETNP (Kienast et al., 2002), suggesting a decoupling
between remineralization rates and N-loss (Ganeshram et al., 2000).

78       While sedimentary $\delta^{15}N$ values are shaped by the sum of N-cycling processes, lipid biomarkers provide more

detailed information (see Rush and Sinninghe Damsté, 2017 for a review). Anammox bacteria biosynthesise $C_{18}$ and $C_{20}$
ladderane fatty acids (FAs) (Fig. 2). These unique lipids contain three or five linearly concatenated cyclobutane rings ([3]-
ladderane and [5]-ladderane, respectively; Sinninghe Damsté et al., 2002). Ladderanes have been successfully applied to
trace abundances of *Ca.* Scalindua spp. in the modern ENP water column (Rush et al., 2012a; Sollai et al., 2015) and as
anammox biomarkers in sedimentary records up to 140 ka (Jaeschke et al., 2009; Rush et al., 2019; van Kemenade et al.,
2023). Moreover, during exposure to oxic conditions ladderane FAs undergo microbially-mediated oxic degradation of the
alkyl side chain by $\beta$-oxidation, in which $C_{18}$- and $C_{20}$-ladderane FAs are sequentially transformed into the short-chain (SC)
$C_{16}$- and $C_{14}$-ladderane partial degradation products (Rush et al., 2011, 2012b). Thus, SC-ladderane FAs in the sediment
archive may be used to trace back anammox cell material that has been exposed to oxic conditions, such as sedimentation
through the oxic water underlying an OMZ. Furthermore, the index of ladderane FAs with five cyclobutane rings ($NL_5$) has
been shown to correlate with the *in situ* water temperature at which ladderane FAs are synthesised (Rattray et al., 2010),
which has been used to determine the provenance of ladderane lipids (Jaeschke et al., 2009; Rush et al., 2012a; Van
Kemenade et al., 2022).



Here, we describe the occurrence of ladderane FAs in a ~160 cal ka BP sediment record from the CCS, covering the
two most recent glacial terminations (T1 and T2). We combined (SC-)ladderanes and the NL$_5$ index with sedimentary bulk
$\delta^{15}N$, stable carbon isotope ratio ($\delta^{13}C$), total organic C (TOC) and total N (TN) to investigate the feedback of changing
OMZ intensity on the occurrence of anammox within the CCS. Moreover, ladderane FAs were also investigated, albeit in
low-resolution, in >160 cal ka BP sediments (up to 500 cal ka BP) to explore their preservation potential.

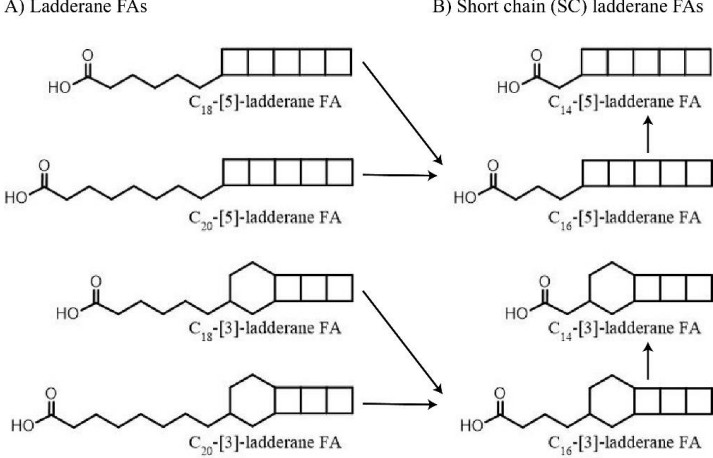


**Figure 1**: Structures of anammox lipid biomarkers used in this study: A) ladderane fatty acids (FAs) with 5 or 3 cyclobutane moieties
containing 18 or 20 carbon atoms. B) short chain ladderane fatty acids (FAs) with 5 or 3 cyclobutane moieties containing 16 or 14 carbon
atoms. Proposed diagenetic pathways are indicated using black arrows (adapted from Rush et al., 2011).
**2 Hydrographic setting**
The northern boundary of the CCS is at the transition zone between the North Pacific Current (NPC) and Alaska gyres
(~50°N) and is bordered in the south by the subtropical waters of Baja California, Mexico (~15–25°N). The CCS (Fig. 2A) is
shaped by: (i) the equatorward California current (CC), extending roughly 1000 km off the North American coast (Checkley
and Barth, 2009), (ii) the poleward, near-shore flowing California undercurrent (CU), and (iii) the seasonal poleward flowing
Davidson current (DC). The CC is a year-round, cold, low-salinity, nutrient-rich surface current (<300 m below sea surface;
mbss), originating from the North Pacific Current.  While the CC is strongest in spring and summer, the DC originating
around Point Conception (35°N) dominates the surface-flow throughout winter. The deeper waters of the CC are shaped by
the NPIW (300–800 mbss), which circulates clockwise in the North Pacific gyre (Sverdrup et al., 1942) and is carried
southwards by the CC. Around Baja California, it convolutes with unventilated intermediate waters of tropical origin, which
have been transported to the eastern Pacific by the Equatorial undercurrent (EUC; Reid, 1997; Reid and Mantyla, 1978).
Here, part of the CC turns north to become the California undercurrent (CU). The CU (~100–300 mbss) carries the warm,



high-salinity, low oxygen waters from Baja California towards Vancouver Island (Thomson and Krassovski, 2010). Within
the CCS, the geostrophic flow of the CC in combination with Ekman transport and eddy activity cause an offshore transport
of (sub-)surface waters and strong coastal jets, which are replaced by the upwelling of the nutrient-rich undercurrent waters
(Huyer, 1983; Chavez and Messié, 2009). Upwelling occurs year-round, and results in high primary production (Bograd et
al., 2009). In the CCS, the high organic matter flux, together with the poor ventilation of the intermediate-water mass (Reid
and Mantyla, 1978; Fu et al., 2018), results in the formation of the ENP OMZ, disassociated in the ETNP (0–25°N; 75–
180°W) and ESTNP (25–52°N; 75–180°W). Dissolved oxygen (DO) concentrations in the cores (<20 µmol kg$^{-1}$) of both the
ETNP (~320–740 meters below sea surface, 'mbss') and ESTNP (~850–1080) OMZ decrease below <1 µmol kg$^{-1}$ (Palmier
and Ruiz-Pino, 2009).

**3 Methods**

**3.1 Sampling location and strategy**

The sediment record was recovered in 1996 during Ocean Drilling Program (ODP) Leg 167 (Lyle et al., 1997) . Site 1012 is
located 105 km offshore California in the East Cortez Basin (32°16.970´N, 118°23.039´W), near the southern front of the
CC and northern front of the ETNP OMZ (Fig. 2B). The core was recovered from a water depth of 1784 m below sea surface
(mbss). For this study, 69 sediment depths (volumes of 20 cm$^3$) were selected for ladderane FAs analysis. Sedimentation
rates ranged from 4 to 15 cm kyr$^{-1}$ (S1, Table 1). Considering the oldest detected ladderane FAs were in 140 ka BP
sediments (~10 m below sea floor 'mbsf') of the Arabian Sea (Jaeschke et al., 2009), we subsampled at a higher resolution
(every 10 to 50 cm) to the first ~160 kyr (15.7 mbsf ) of the record (with a maximum resolution of 10 cm around T1 and T2)
and at a lower resolution (80 to 200 cm) to ~500 cal ka BP (37.5 mbsf). In addition, 74 sediments (10-50 cm resolution) were
analysed for bulk sedimentary organic carbon (TOC) and N (TN) content, and bulk isotopic ratio values (δ$^{15}$N and δ$^{13}$C). A
detailed overview of all samples is given in Supplement 1, Tables 1 and 2. Samples were freeze-dried and stored at -20 °C
prior to analysis.

**3.2 Analysis of sedimentary bulk TOC, TN, δ$^{13}$C and δ$^{15}$N**

Sediments were freeze-dried and ground to powder. For TOC and δ$^{13}$C analysis, aliquots of bulk sediment were decalcified
to remove all carbonates. Samples were first acidified with 2M hydrochloric acid (HCl) and rinsed with distilled water to
remove the salts. After the decalcification step, ca. 0.5 mg of dried material was used for the analysis. For TN and stable
nitrogen isotope ratio (δ$^{15}$N) between 15 and 20 mg of non-decalcified sediment were used. All samples were packed in tin
cups and introduced to the Thermo Scientific Flash 2000 elemental analyzer coupled to a Thermo Scientific Delta V
Advantage isotope ratio mass spectrometer (EA/IRMS). Results are expressed in standard notation relative to Vienna Pee



Dee Belemnite (VPDB) for $\delta^{13}$C and relative to air for $\delta^{15}$N. The precision as determined using laboratory standards
calibrated to certified international reference standards was in all cases < 0.2 ‰.

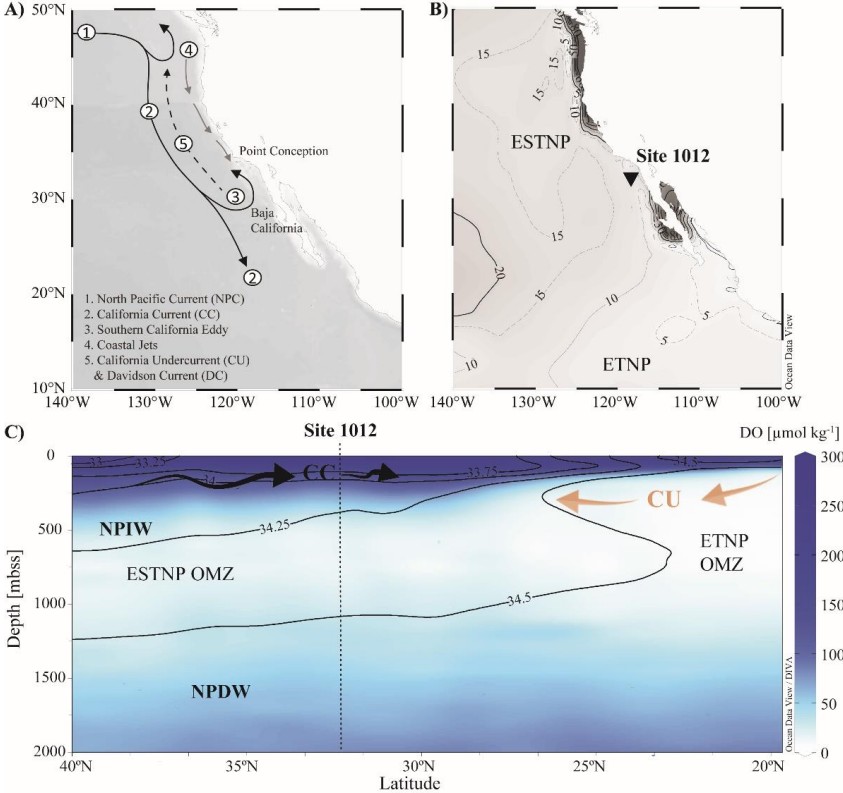

**Figure 2:** A) map of the California Current System (CCS). Key currents are indicated with arrows. B) location of ODP site 1012
(32°16.970´N; 118°23.039´W) recovered at 1784 mbss, with minimum dissolved oxygen (DO) concentrations [µmol kg⁻¹] detected in the
water column in 2018 (WOA, 2018). C) A latitudinal section plot of the CCS water column showing modern annually averaged DO (µmol
kg⁻¹) concentrations and salinity (psu) concentrations with the color bar and contour lines, respectively (WOA, 2018). Major current and
water masses are also indicated, i.e., the Eastern Tropical and Eastern Subtropical North Pacific (ETNP and ESTNP, respectively) OMZs,
the California Current (CC; black arrows), the California Undercurrent (CU; orange arrows), North Pacific intermediate waters (NPIW)
and North Pacific deep water (NPDW). Maps were created in Ocean Data View and DIVA gridding was applied for interpolation of DO
concentrations (Schlitzer and Reiner, Ocean Data View, 2021).
**3.3 Age model**
Liu et al. (2005) previously constructed an age model for ODP site 1012, based on sediments recovered from Hole B. As the
material used in this study is predominantly from Hole A and C, a revised age model was constructed (S1, Table 1). The
revised age model for sediments up to 160 cal ka BP (15.7 m composite depth, 'mcd') was created by correlation of the bulk



sedimentary $\delta^{15}$N record of Liu et al., (2005) with our dataset. Tie points (age *vs* composite depth) were selected by fine-
tuning using QAnalyseries (version 2022). For sediments >160 cal ka BP, which were solely sampled for ladderane FAs at
low resolution (i.e. not sedimentary $\delta^{15}$N), the age model of Liu et al. (2005) is used.

**3.4 Ladderane extraction**

Homogenized, freeze-dried sediments were extracted using a low temperature - low pressure accelerated solvent extraction
(ASE) method, previously described for ladderane extraction in Rush et al. (2012b). Thereafter, aliquots of the total lipid
extract were saponified in 2 N potassium hydroxide (in a 96 % MeOH solution) by refluxing for 1 h. After, 2 mL of
bidistilled water was added. The saponified extracts were acidified by adjusting the pH to 3 with 2 N hydrochloric acid (in a
50 % MeOH solution). Phase separation was induced by adding 2 mL of DCM. The biphasic mixtures were sonicated for 5
min and centrifuged for 2 min (3000 rpm). The DCM layers, containing the FAs, were collected. The mixtures were
partitioned twice more with DCM, after which the same procedure was applied before collection of the DCM layers. The FA
fractions were dried over a sodium sulphate ($Na_2SO_4$) column. Then, the fractions were methylated with diazomethane to
convert FAs into their corresponding fatty acid methyl esters (FAMEs) and allowed to airdry overnight to avoid losing the
more volatile SC-ladderane FA had they been dried under a stream of $N_2$. The methyl esters of the polyunsaturated fatty
acids (PUFAs) were removed by eluting the FAME fractions with DCM over a silica impregnated silver nitrate ($AgNO_3$)
column. FAME fractions were dissolved in acetone and filtered over 0.45 mm PTFE filters (4 mm; BGB, USA).

**3.5 Ladderane analysis**

A commercially available deuterated $C_{20}[5]$-PUFA (Reagecon Diagnostics Ltd.) was added as an internal standard to the
FAME fractions. FAME fractions were analysed on an Agilent 1290 Infinity I ultra-high performance liquid
chromatographer (UHPLC), equipped with a thermostatted auto-injector and column oven, coupled to a Q Exactive Plus
Orbitrap MS, with an atmospheric pressure chemical ionization (APCI) probe (Thermo Fischer Scientific, Waltham, MA)
operated in positive ion mode. Separation was achieved with a ZORBAX Eclipse XDB $C_{18}$ column (Agilent, 3.0×250 mm,
5 μm), using MeOH as an eluant (0.4 ml min$^{-1}$). APCI source settings were set as follows: corona discharge current, 2.5 μA;
source CID, 10 eV; vaporizer temperature, 475℃; sheath gas flow rate, 50 arbitrary units (AU); auxiliary gas flow rate,
30AU; capillary temperature, 300℃; and S-lens, 50V (van Kemenade et al., 2022). A mass range of *m/z* 225–380 was
monitored (resolution 140,000 ppm), followed by data-dependent MS$^2$ (resolution 17,500 ppm at *m/z* 200), in which the 10
most abundant masses in the mass spectrum were fragmented successively (stepped normalised collision energy 20, 25, 30).
An inclusion list containing the exact masses of $C_{14-24}$-[3]- and $C_{14-24}$-[5]-ladderane FAMEs was used. Mass chromatograms
(within 5 ppm mass accuracy) of the protonated molecules ([M+H]$^+$) were used to integrate the detected ladderanes: $C_{14}[3]$-,
$C_{14}[5]$, $C_{16}[5]$, $C_{18}[3]$-, $C_{18}[5]$-, $C_{20}[3]$- and $C_{20}[5]$-ladderane FAMEs (*m/z* 235.169, 233.154, 261.185, 291.232, 289.216,



319.263 and 317.248, respectively), and the internal deuterated $C_{20}[5]$-PUFA standard ($m/z$ 322.279) . Identification of
ladderanes was achieved by comparing retention times and spectra with in-house isolated $C_{20}[3]$- and $C_{20}[5]$-ladderane
FAME standards (Hopmans et al., 2006; Rattray et al., 2008) and with ladderane FAMEs in a biomass sample of *Ca.*
Kuenenia.

191        Previously, ladderane FAME quantification has been conducted using calibration curves of in-house isolated $C_{20}[3]$-

and [5]-ladderane standard (Hopmans et al., 2006). However, this quantification method does not correct for any variability
in ion intensity, due to e.g., matrix effects and/or changes in the instruments functioning.  Therefore, we further optimised
this quantification method to include a response correction using a commercially available internal standard (deuterated
$C_{20}[5]$-PUFA). At the start of each sequence, calibration curves were made for the $C_{20}[3]$- and [5]-ladderane standards *and*
the deuterated $C_{20}[5]$-PUFA standard. The relative response of the deuterated $C_{20}[5]$-PUFA commercial standard in relation
to the ladderane FAME standards was determined from the slopes of their calibration curves (giving a relative response
factor, i.e. RRF). An RRF of 1.3 was used for [3]-ladderanes, based on the $C_{20}[3]$-ladderane, and an RRF of 1.2 for the [5]-
ladderane, based on  the $C_{20}[5]$-ladderane. Using the RRFs, ladderane FAME concentrations ($C_L$, expressed in µg · g dry
weight$^{-1}$) were calculated as follows:
$$C_L = \frac{m_{IS}\left(\frac{A_L}{\left(\frac{A_{IS}}{RRF}\right)}\right)}{m_S} \qquad [1]$$

With $m_{IS}$ being the mass (µg) of the added internal standard, $m_S$ the dry weigh of extracted sediment (g), $A_L$ the integrated
peak area of the given ladderane FAME , $A_{IS}$ the integrated peak area of the internal standard, and RRF the relative response
factor. Ladderane concentrations (including concentrations normalized against gram TOC) are reported in supplement 1
(Tables 4 and 5). To compare with previous studies that did not use an internal standard, the established method that uses
external calibration curves of three authentic standards (Hopmans et al., 2006; Rush et al., 2012b; Rattray et al., 2010) was
also performed (S1, Table 8b; S2.2).
**3.6 NL$_5$ index**
The index of ladderane lipids with five cyclobutane rings (NL$_5$) correlates with the temperature at which they were
synthesised. The NL$_5$ index is calculated according to the following equation:
$$NL_5 = \frac{C_{20}[5]\text{ladderane FA}}{C_{18}[5]\text{ladderane FA}+ C_{20}[5]\text{ladderane FA}} \qquad [2]$$

The empirical fourth-order sigmoidal relationship between the NL$_5$ index and temperature is then described by:



$$NL_5 = 0.2 + \frac{0.7}{1 + e^{-(\frac{T-16.3}{1.5})}}$$ [3]
with temperature (T) in °C (Rattray et al., 2010).

**3.7 Degradation rates and constants**

Ladderane degradation rates were calculated using the following equations for lipid degradation constants and rates (Canuel
and Martens, 1996):
$$k' = \frac{-\ln [\frac{C_t}{C_{t0}}]}{t}$$ [4]
With $k'$ being the first order rate constant (kyr$^{-1}$), $C$ being the concentration (µg g sediment$^{-1}$) at time $t$ ($C_t$) and at the initial
time ($C_{t0}$), and $t$ being the relative time (kyr).

**4 Results**

**4.1 Bulk sedimentary total nitrogen and total organic carbon**

Bulk sedimentary total nitrogen (TN) ranges between 0.1–0.6 % throughout the record. $\delta^{15}N$ fluctuates from 5.8 to 10.0 ‰.
An offset of 3 to 4 ‰ is observed between interglacials and glacials, with higher values during interglacials. The content of
sedimentary total organic carbon (TOC) varies between 1.7–7.4 % throughout the record, whilst its carbon isotopic
composition ($\delta^{13}C_{TOC}$) ranges from -23.0 to -21.6 ‰ (S1, Table 3).

**4.2 Ladderane FAs**

**4.2.1 Ladderane FAs concentrations & the NL$_5$ index**

The ladderane fatty acids identified in this record are C$_{18}$[5]-, C$_{18}$[3]-, C$_{20}$[5]- and C$_{20}$[3]-ladderanes and their diagenetic
products, the SC C$_{14}$[5]-, C$_{14}$[3]- and C$_{16}$[5]-ladderanes. Normalized concentrations over the 160 ka record ranged as follows:
C$_{14}$[5]-ladderane 16–158 ng gTOC$^{-1}$, C$_{14}$[3]-ladderane 27–184 ng gTOC$^{-1}$, C$_{16}$[5]-ladderane 34–198 ng gTOC$^{-1}$, C$_{18}$[5]-
ladderane 7–107 ng gTOC$^{-1}$, C$_{18}$[3]-ladderane 4–76 ng gTOC$^{-1}$, C$_{20}$[5]-ladderane 5–79 ng gTOC$^{-1}$, and C$_{20}$[3]-ladderane 10–
208 ng gTOC$^{-1}$ (S1, Table 4). Summed SC-ladderane and ladderane concentrations over the entire 500 ka record are 0.5–33
and 0.1–23 ng g$^{-1}$ dry weight, respectively (Fig. 3; S1 Table 5). Concentrations calculated without the use of the internal
standard (Hopmans et al., 2006; see section 2.5), are reported in S1 (Table 8b). The NL$_5$ index (eq. [2]) ranges from 0.3 to
0.8 throughout the record. Corresponding NL$_5$-derived temperatures (eq. [3]) are between 13.1–18.6°C (S1, Table 6).



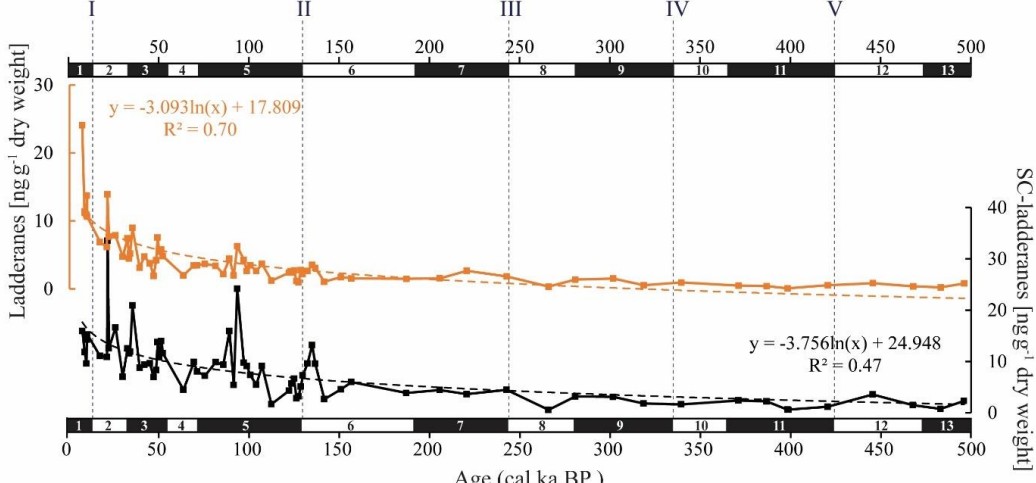

**Figure 3:** Summed $C_{18}[5]$-, $C_{18}[3]$-, $C_{20}[5]$- and $C_{20}[3]$-ladderane (orange) and summed short-chain (SC) $C_{14}[5]$-, $C_{14}[3]$- and $C_{16}[5]$-ladderane (black) concentrations (ng g$^{-1}$ dry weight) in the ODP 1012 record. The logarithmic relationship between ladderanes and SC-ladderanes with time is provided (with corresponding $R^2$), and displayed with orange and black spaced lines, respectively. Grey spaced lines indicate the approximate timing of glacial terminations I to V. N.B. the scales of the y-axes are different.

## 5 Discussion

In the sediment record of ODP site 1012, both short chain $C_{14}[3]$-, $C_{14}[5]$-, $C_{16}[5]$- as well as $C_{18}[3]$-, $C_{18}[5]$-, $C_{20}[3]$- and $C_{20}[5]$- ladderane FAs were detected over the last 500 kyr (~38 mbsf; Fig. 3). This poses a considerable extension of the ladderane record (formerly detected up to ~140 ka BP in Arabian Sea sediments; ~10 mbsf; Jaeschke et al., 2009). Below, we will first discuss the provenance of the detected ladderane lipids (section 5.1). Then, their variability throughout glacial-interglacial cycling (section 5.2), ending with the subsequent implications on our understanding of the nitrogen cycle of the CCS (section 5.3). Unfortunately, the coarse sampling resolution in >160 cal ka BP sediments may have missed important variations, and therefore, analysis of trends in ladderane concentrations over (inter)glacial cycling is limited to <160 cal ka BP sediments.

### 5.1 Ladderanes sourced from anammox bacteria in the ESTNP OMZ water column

The relative contribution of SC-ladderanes to the total ladderane pool is a measure of oxygen exposure (Rush et al., 2011, 2012b), and the $NL_5$-index is a measure of the water temperature of the niche of anammox bacteria (Rattray et al., 2010). In combination, these data may provide insights into the origin of ladderanes in the CCS sediment record.



**Figure 4:** From top to bottom: total ladderane (summed SC-ladderane and ladderane) normalized against TOC [ng gTOC⁻¹], ladderane and SC-ladderane accumulation rates [µg⁻¹ cm⁻² kyr⁻¹], relative abundance of SC-ladderanes over total ladderanes [%], total nitrogen (TN) from Liu et al., (2005) and this study [%], bulk sedimentary $\delta^{15}N$ from Liu et al., (2005) and this study [‰], $U^{K'}_{37}$ derived sea-surface temperatures (SST) from Herbert et al., (2001) and NL₅-derived temperatures from this study [°C], benthic $\delta^{18}O$ record from Herbert et al., (2001) [‰], total organic carbon (TOC) [%] and bulk sedimentary $\delta^{13}C$ [‰]. All data is derived from the same location (ODP site 1012).





Marine isotope stages (MIS) are indicated with black and white bars. Periods of maximum global ice volume (Herbert et al., 2001; blue
bars), deglaciation (striped blue bars) and the approximate timing of glacial terminations TI and TII (dashed lines) are also indicated.
In the CCS, a progressive depletion of both the water column $\delta^{15}N_{NO3}$ and sedimentary $\delta^{15}N$ signal occurs with
increasing latitude, resulting in more depleted values at ODP site 1012 (8–10 ‰; Altabet et al., 1999; Liu et al., 2005; this
study) than in the ETNP OMZ core. The northward transport of denitrified waters by the poleward flowing oxygen-poor CU
from the core of the ETNP has been evoked to explain this trend (Castro et al., 2001; Kienast et al., 2002). However, a
similar mechanism is unlikely to explain the presence of ladderane FAs at ODP site 1012. Ladderane FAs are relatively
labile compounds, and in the Arabian Sea have been shown to already degrade into their SC-products (at relative proportions
of ~20 %) within the OMZ water column (DO <3 µmol L$^{-1}$). There, the sinking of ladderanes through the oxygenated
bottom waters underlying the OMZ ultimately resulted in a relative abundance SC-ladderanes in the surface sediments of
20–80 %, depending on water column depth (Rush et al., 2012b).
At ODP site 1012, SC-ladderanes were present in similar relative abundances (40–88 %) throughout the record
(Fig. 4). The similarly high contribution of SC-ladderanes in the ODP 1012 record suggest ladderanes are also sourced from
an overlying OMZ water column (i.e. the ESTNP OMZ) and sunk through oxygenated bottom waters before being deposited
on the seafloor, which readily became anoxic in view of the high TOC content (Fig. 4). An OMZ water column source is
consistent with NL$_5$-derived temperatures (13–17°C; S1, Table 6), which are significantly higher than what would be
expected for sedimentary anammox bacteria (i.e., modern annual average bottom water temperatures at site 1012 are <5°C;
WOA, 2018). And, while transport of ladderane FAs has been shown to occur within oxygen-depleted systems (van
Kemenade et al., 2022), long-distance transport of ladderane FAs with the CU (characteristic DO concentration of ~62 µmol
L$^{-1}$ in modern CU water; Sahu et al., 2022) is unlikely, and would be expected to yield higher relative abundances of SC-
ladderane FAs than detected in the record. Transport of ladderanes is also not reflected in present-day ENP ladderane
distributions, as an investigation of ladderanes at a more northerly (~20°N) and a more southerly (~17°N) located site
showed *in situ* synthesis by pelagic *Ca.* Scalindua at both sites (Sollai et al., 2015). Hence, ladderane FAs are thought to
predominantly derive from the ESTNP OMZ water column and reflect a local anammox signal.
**5.2 Anammox variability in the CCS over the last 160 kyr**
**5.2.1 The Holocene and MIS-5, including the penultimate interglacial of MIS 5e**
Over the ~500 cal ka BP record, ladderane FAs are observed to decrease logarithmically with time (Fig. 3; $R^2 = 0.70$), in
which the degradation constant $k$ follows a linear relationship (when logarithmically transformed; Fig. 5A; $R^2 = 0.88$) with
time. This is consistent with first order degradation kinetics, typical for OM (Canuel and Martens, 1996). As such, it is not
surprising that the highest ladderane concentrations are observed in the youngest sediments, deposited during the early to



mid-Holocene. Even so, ladderane FAs normalized against TOC also show elevated concentrations in Holocene sediments.
This suggests high ladderane FAs at this time are not simply a preservation signal but also reflect an increase (compared to
pre-Holocene sediments) in their production by *Ca.* Scalindua spp. relative to the total organic C pool. Moreover, elevated
ladderanes in early to mid-Holocene sediments coincide with enriched bulk $\delta^{15}N$ (9–10 ‰), indicative of increased N-loss by
anaerobic microorganisms, and TOC and TN concentrations (Fig. 4), indicative of increased productivity.

296   Interestingly, SC-ladderane FA concentrations are not highest in Holocene sediments. Consequently, the

SC-ladderane data does not fit the logarithmic decrease with time well ($R^2 = 0.34$; Fig. X), which is also reflected in the
relationship of the degradation constant $k$ with time (Fig. 5A; $R^2 = 0.43$). The oxidation of ladderane FAs to produce SC-
ladderane FAs (Rush et al., 2011) has been shown to take place within the oxic waters below the OMZ. In this way, 20–80 %
of the ladderane FAs were transformed into SC-ladderanes in the Arabian Sea (Rush et al., 2012c). Throughout the deeper
CCS sedimentary record (>10 cal ka BP), the relationship between ladderane FAs and their SC-products follows a linear
trend ($R^2 = 0.88$; Fig. 5B), with SC-ladderanes making ~60–80 % of total ladderanes (Fig. 4). However, in Holocene
sediments (<10 cal ka BP sediments), the relationship between ladderanes and SC-ladderanes is different (Fig. 5B), and SC-
ladderanes occur at relatively lower abundance (40– 60 % in Fig. 4) compared to the rest of the record. This appears to
indicate that after 10 cal ka BP, there was no significant change in the exposure of ladderane FAs to the oxygenated water
underlying the ETNP OMZ before being buried in the sediment record, but that in the recent record, there was reduced
oxygen exposure.

308   Lembke-Jene et al. (2018) showed, using palaeoceanographic proxies and palaeomodeling, that a combination of

sea ice loss, increased SST and remineralization rates led to more deoxygenated intermediate waters (the NPIW) during the
early to mid-Holocene in the North Pacific. Moreover, in the ETNP, enriched sedimentary $\delta^{15}N$ values and laminated
sediments during the early Holocene, alongside geochemical tracers, have been interpreted to signal the presence of a strong
OMZ at this time, while bioturbated sediments occurred over the last glacial period (Thunell and Kepple, 2004).

313   Ladderane FAs concentrations also peak during the penultimate interglacial (the Eemian; MIS 5e), in line with

enriched (>8 ‰) $\delta^{15}N$ values. Microfossil data from MIS 5 has indicated that intermediate waters in the western North
Pacific were more deoxygenated during the Eemian (Matul et al., 2016), which may have driven increased anammox in the
CCS at this time. However, ladderane FAs concentrations during mid-MIS 5 (MIS 5b–c; Fig. 4) are even more elevated,
while the $\delta^{15}N$ signal here is subdued (<8 ‰). Ladderane trends in MIS 5 hereby seem to follow paleo-productivity proxies
(i.e., TOC and TN) more closely (which also peak during MIS 5b–c; Fig. 4). During MIS 5b–d, intermediate waters in the
western North Pacific were oxic (Matul et al., 2016). Indeed, over the course of MIS 5, from late MIS 5e onwards, SSTs in
the CCS decreased while the CC strengthened (Herbert et al., 2001; Yamamoto et al., 2007). This would have led to



increased transport of high-oxygen, nutrient-rich NPIW (Herguera et al., 2010) and enhanced open ocean upwelling. At the
same time, this would have fuelled productivity, which is reflected in the high TOC and TN concentrations in mid-MIS 5.

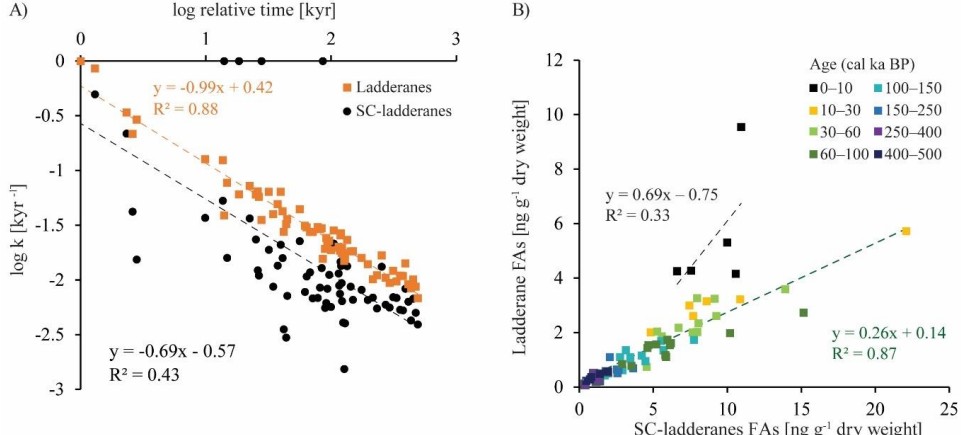


**Figure 5:** A) Linear relationship between the logarithmic values of the degradation constant k and relative time for ladderane FAs (orange

squares) and SC-ladderane FAs (black dots). B) Relationship between ladderane FAs and SC-ladderane FAs, in which samples are colour-
coded according to age. The linear relationship and corresponding $R^2$ are given for the most recent age group (0-10 cal ka BP; in black)
and the >10 cal ka BP age groups (in green).
Babbin et al., (2014) showed, using incubations from the ETNP OMZ, that anammox rates increase in response to
the addition of OM. Likewise, in the modern Southern Pacific OMZ, N-loss by anammox was found to be strongly
correlated with the export of OM, via the release of ammonium into the water column through remineralization (Kalvelage et
al., 2013). As such, the co-variation of ladderane FAs with paleo-productivity proxies, could reflect an increase in *Ca.*
Scalindua spp. abundance in response to an increased N-substrate supply via OM-remineralization or nutrient transport,
rather than a response to changing DO concentration. Even so, the increased OM-supply during MIS 5b–d could also have
led to more reducing conditions via OM-remineralization within the ENP OMZ, which may not be recorded in the western
part of the North Pacific. The discrepancies between the ladderane- and $\delta^{15}N$ record at this time, and consequent implications
for our understanding of the N-cycle in the CCS are further discussed in section 5.3.
**5.2.2 The two most recent glacial periods**
Ladderane FAs are observed to increase from early MIS 3 to mid-MIS 2, and from mid- to late-MIS 6. Maxima of
ladderanes occur approximately at the timing of icesheet volume maxima of the last glacial maxima (LGM) and the
penultimate glacial of MIS 6 (blue bars in Fig. 4; following timing of Herbert et al., 2001). During the last glacial period
(~115–12 ka BP) and the penultimate glacial MIS 6, large parts of the North American continent were covered by the
Laurentide and Cordilleran ice sheets. While glacials are typically associated with a well-ventilated intermediate-water mass





(Herguera et al., 2010) and a strong southward advection of the CC (Ortiz et al., 1997), a weakening of the CC has been
proposed to occur at times of global ice sheet maxima. In the CCS, $U^{K'}_{37}$-derived temperatures indicate that SSTs increased
~12 kyr in advance of maximal ice-sheet volumes. This is thought to reflect increased northward advection of warm oxygen-
poor waters carried by the CU and DC in response to a weakened CC due to large ice-sheet volumes (Herbert et al., 2001).
Using trace elements, Cartapanis et al., (2011) found that intermediate water oxygenation off Baja California deteriorated
slightly over the course of late MIS 3 and early MIS 2, consistent with a strengthening of the CU at this time. As such, the
increased abundance of ladderanes observed during (and leading up to) ice sheet maxima at ODP site 1012, may derive from
an increased *Ca.* Scalindua spp. abundance due to more reduced local conditions, via the enhanced strength of the CU.

351        In contrast, low ladderane concentrations occur at times of deglaciation (T1: ~19–11 ka BP and T2: ~135-128 ka

BP). Modelling-studies have proposed that during the early part of the Last Glacial Termination (~17.5–15.0 ka BP), a
reorganization of the global conveyor belt circulation would have led to deep water formation in the North Pacific, extending
to ~2500 to 3000 mbss. In turn, this would have led to nutrient-poor but well-ventilated intermediate-deep waters (Okazaki
et al., 2010; Menviel et al., 2011).  At the same time, there was increased influx of freshwater from the Cordilleran ice-sheet
into the northeastern Pacific. Increased melt-water influx during deglaciation would have strengthened the southward forcing
of the oxygen-rich CC, and consequently weakened the CU (Herbert et al., 2001). As such, increased ventilation of glacial
NPIW and decreased northward forcing of the CU may have reduced the extension of the anoxic ESTNP core available for
anammox, which may explain the observed ladderane minima during deglaciation.  Ladderane and $\delta^{15}$N minima (~5.9 ‰
during T1 and ~6.8 ‰ during T2) coincide, suggesting limited loss of bioavailable N via anammox and denitrification at this
time. In contrast, in the Gulf of Tehuantepec Thunell and Keppel (2004) recorded increasing $\delta^{15}$N values over 23–17 ka, with
maximum values during the Bølling–Allerød warming period. Differences between the $\delta^{15}$N records between this (~15°N)
and more northerly located sites (e.g., ODP site 1012) over T1, has been explained by the presence of a hydrographic
boundary within the ETNP around ~20°N at this time, which kept northern- and southern-sourced intermediate waters
separate (Hendy and Pedersen, 2006).
**5.3 Implications of the occurrence of anammox on the N cycle in the CCS**
In the CCS, previous estimates of changes in N-loss over time have been based on the bulk sedimentary $\delta^{15}$N record.
Enriched $\delta^{15}$N during interglacials (7–10 %) are thought to reflect intensified denitrification in response to reduced DO,
while more depleted $\delta^{15}$N during glacials (4–6 %) are assumed to reflect lowered rates in response to increased DO (Liu et
al., 2005; 2008). However, the high abundance of ladderane FAs throughout our CCS record (i.e. up to a factor ~5 higher
than in the Arabian Sea record; Jaeschke et al., 2009) now shows that anammox was (also) responsible for N-loss and thus
contributed, at least partially, to the sedimentary $\delta^{15}$N record.



The correlation of the $\delta^{15}$N record with SST reconstructions (Liu et al., 2005) shows that fluctuations in $\delta^{15}$N occur
in tandem with glacial-interglacial cycling. However, a long-standing conundrum has been the discrepancy between the $\delta^{15}$N
record and productivity proxies (i.e., TOC and TN), especially north of the ETNP (Kienast et al., 2002), as also seen in our
record (Fig. 4). This decoupling has been used previously to suggest that variations in denitrification was not due to changes
in OM remineralization rate, but rather from changes in ocean circulation and ventilation patterns (Ganeshram et al., 2000).
Yet, fluctuations in ladderanes *do* seem to follow trends in paleo-productivity proxies (i.e., TOC and TN) relatively closely,
especially during the Holocene, MIS 3 and MIS 5. And, while enriched $\delta^{15}$N values sometimes correspond to ladderane
maxima (i.e. during the Holocene), discrepancies with ladderane concentrations are seen especially during MIS 3 and MIS 5,
and during glacial periods (Fig. 4).
Out-of-phase anammox and denitrification could be caused by variations in the C:N ratio of OM. Given the average
C:N signature of marine OM (106:16; Redfield, 1963), stoichiometric constraints should result in a ratio of $N_2$ production via
denitrification and anammox of 71:29 (Koeve and Kähler, 2010). Localized variations in the C:N signature may result in
different relative contributions. Yet, integrating these variations over space and time should obtain a similar ratio (Dalsgaard
et al., 2012; Ward, 2013; Babbin et al., 2014). As such, given the temporal resolution of the record (which does not cover
seasonality), denitrification and anammox intensities are expected to fluctuate in-tandem.
Moreover, both denitrifiers and anammox bacteria are similarly inhibited by oxygen in the marine environment, at
DO concentration above 3 to 8 µmol L$^{-1}$ (Babbin et al., 2014). Furthermore, Babbin et al., (2014) showed, using incubations
from the ETNP OMZ, that both denitrification and anammox are limited by OM supply, and their rates increase in response
to the addition of OM. As anammox bacteria are autotrophic, this may be explained by the dependence of the process on
$NH_4^+$ and $NO_2^-$ availability, which can a.o. be supplied via remineralization. As such, both anammox and denitrification
should respond similarly to changes in DO and OM in the CCS.
Reconstructions of N-loss using sedimentary $\delta^{15}$N depend on the assumption that there was complete biological
utilization of $NO_3^-$ by phytoplankton. However, during periods of high upwelling intensity (as likely occurred during mid-
MIS 5; see section 5.2.1), the high $NO_3^-$ availability may result in incomplete $NO_3^-$ assimilation. This allows for the
preferential uptake of $^{14}$N by primary producers, resulting in a pool of $\delta^{15}$N depleted OM available for heterotrophic
denitrification (Tesdal et al., 2013). Hence, at times of high $NO_3^-$ supply, incomplete nitrate assimilation would have
quenched the $\delta^{15}$N signal, even if denitrification was as intense as during periods of low $NO_3^-$ availability. Moreover, a study
by Altabet and Francois (1994) showed that sedimentary $\delta^{15}$N in the equatorial Pacific records the isotopic enrichment of
near-surface $NO_3^-$ via depletion by phytoplankton, in which enriched $\delta^{15}$N values are associated with reduced $NO_3$
availability for phytoplankton assimilation. Also, in the South Pacific, $NO_3^-$ concentrations have been found to affect the U
$^{K'}_{37}$ index (Placencia et al., 2010). Given the excellent correlation between the $\delta^{15}$N and U $^{K'}_{37}$ -based SST records of the CCS



(Liu et al., 2005) and the discrepancies between the $\delta^{15}N$ and ladderane records, it may be sensible to conclude that the CCS
sedimentary $\delta^{15}N$ fluctuations (also) record variations in $NO_3^-$ assimilation by phytoplankton.
Additionally, other biological processes may influence the $\delta^{15}N$ signal (Zonneveld et al., 2010). In the Gulf of
Tehuantepec, at the southern end of the ETNP OMZ core, $\delta^{15}N$ values decrease over the course of the Holocene (Thunell and
Kepple, 2004; Hendy and Pedersen, 2006), while laminated sediments suggest reduced DO concentrations. This was
interpreted as being the result of increased $N_2$-fixation ($\varepsilon: \leq +2$ ‰; Sigman and Fripiat, 2019), which lowered the
"denitrification" $\delta^{15}N$ signal (Thunell and Kepple, 2004). Lastly, enrichment of the sedimentary $\delta^{15}N$ values occurs during
early burial, where oxygen exposure results in enhanced biological isotopic alteration (Robinson et al., 2012). In short,
sedimentary $\delta^{15}N$ is shaped by many opposing processes, and assuming a one-on-one relationship with denitrification
intensities and DO concentration clearly misses the complexity that shape the CC system. Ladderanes hereby offer a more
detailed picture of N-loss dynamics in the paleoenvironment of the CCS. In the case of the ODP site 1012 record, ladderane
concentration trends challenge the conventional assumption that N-loss processes solely follow ocean circulation and
ventilation patterns coupled to (inter)glacial cycling, and instead show OM remineralization may also be an important driver
of N-loss.
Discrepancies between the ladderane and $\delta^{15}N$-record hereby necessitate careful consideration when applying N-
isotope based budgets to estimate past N-cycling. More specifically, the occurrence of increased ladderane concentrations
during glacial maxima may require a re-evaluation on the response of N-loss rates to glacial-interglacial cycling in the CCS.
Furthermore, the occurrence of an additional N-loss pathway in the CCS (anammox), other than denitrification, may affect
estimates of $N_2O$ greenhouse-gas production by denitrifiers and the degree of heterotrophy of the system, although the
importance of this would require further investigation. Future research, investigating anammox biomarkers in other CCS
records (preferably in a latitudinal gradient with this record) may offer further insights into N-loss dynamics across glacial-
interglacial cycles.
**6 Conclusion**
Ladderane FAs detected in a ~500 kyr CCS sedimentary record at ODP site 1012 reveal the past occurrence of anaerobic
ammonium oxidising (anammox) bacteria in the water column of the California current system (CCS) over the last five
glacial terminations. The index of ladderanes with five cyclobutene moieties ($NL_5$), which correlates with the *in situ*
temperature at which ladderanes are synthesised, suggests that ladderanes were derived from the ETSNP OMZ water
column. The high-resolution record of the last two interglacial-glacial transitions shows a continuous presence of ladderane
FAs, with maxima during: i) the Holocene, ii) leading up to and during the LGM (early MIS 3 to mid-MIS 2), iii) MIS 5b-c
and iv) during the ice sheet maxima of the penultimate glacial (late MIS 6). Combining information on the presence of



ladderanes with paleo-productivity proxies and the hydrographic features of the CCS suggests anammox abundance was
driven both by OM-remineralization and advection changes, which regulated nutrient and oxygen concentrations. In the
record, a clear shift is seen in the relationship of SC-ladderanes over their parent products, in which the relative abundance of
SC-ladderanes is significantly lower in Holocene than in pre-Holocene sediments. This may reflect a shift in oxygen
exposure, which corresponds to previous studies showcasing a vertical expansion of the ENP OMZ over the Holocene.
Clearly, the anammox contribution to N-loss in the CCS, as shown in this study, requires a reassessment of biogeochemical
cycling in this system. Discrepancies between the ladderane and $\delta^{15}$N record may imply that N-loss was perhaps more
intense during cold phases than previously assumed. Careful considerations must thus be taken when using N-isotope based
budgets to estimate past N-cycling in the CCS; sedimentary $\delta^{15}$N is shaped by many opposing processes, and assuming a
one-to-one relationship between N-loss intensities and OMZ variability clearly overlooks the complexity that shapes the CC
system. Ladderanes hereby offer a more holistic picture of N-loss dynamics in the paleoenvironment of the CCS.
**Data availability.** All data discussed in this paper is available in the supplementary material 1. Data from supplementary
material 1 can be retrieved via the following doi: 10.25850/nioz/7b.b.sg
**Supplement.** The supplement related to this article is available on-line at:
**Author contributions.** ZE and ZRvK performed the laboratory work. ZRvK conducted the data analysis and writing of the
manuscript. ZE created the age-model. ECH developed and optimized the UHPLC-HRMS method for the analysis of
ladderane lipids. DR provided the supervision of the project. DR, ZE and ZRvK designed and conceptualized the project.
JSDD provided critical support in data interpretation. All authors contributed to the writing of the manuscript.
**Competing interests.** The authors declare that they have no conflict of interest.
**Acknowledgements.** We thank the captain and crew of Ocean Drilling Program Leg 167 for the collection of all sampled
material used in this study. Denise Dorhout and Monique Verweij are also greatly appreciated for their support with the
instrumental analysis. We also kindly thank Ronald van Bommel and Marcel van der Meer for their help in the isotope lab.
**Financial support.** This research was supported by the Soehngen Institute of Anaerobic Microbiology (SIAM) Gravitation
Grant (024.002.002), awarded to JSSD by the Dutch Ministry of Education, Culture and Science (OCW).

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
