# Peer review of "Loss of nitrogen via anaerobic ammonium oxidation (anammox) in"

_EGUsphere, 2023_

## Author Response (AR1)

We greatly appreciate the time and effort the Associate editor and reviewers have taken to comment on our manuscript to improve its content. We have adjusted the manuscript accordingly. Please find below our response to individual comments in bold. Reference to line numbers refer to the revised manuscript .pdf file, without tracked changes.

On behalf of all co-authors,
Zoë van Kemenade

*Associate editor:*

*The reviewers have raised several good questions and provided thoughtful comments to further improve the manuscript. Thank you for your responses. The reviewer comments also covered several questions that I had during my own review.*

*My only additional thoughts related to the reviewer comments and your responses are:*
*- It would be simple to refer to Figs 3 and 4 together at the first time, in which case you do not run into problems with the order of figures regarding the suggestions from reviewer 1.*
*- Specifically to the comments from reviewer 2, I actually agree that likely only the earlier part of the record is a reliable reconstruction, but the older part may be largely affected by degradation of ladderanes. But it is certainly a possibility that contributions from anammox are much lower relative to denitrification. So I certainly think you should discuss these possibilities, in agreement with the related reviewer comments and your responses.*

*I also agree with the reviewers that minor revision is required and I look forward to your revised manuscript that includes the feedback received from the reviewers as per your responses.*

*Thank you.*

*Kind regards,*
*Dr Sebastian Naeher*
*Associate Editor, Biogeosciences*

We thank Dr. Sebastian Naeher for taking the time to evaluate our manuscript and greatly appreciate the feedback. We agree with the Associate editor's comments that the older part of the record is largely affected by degradation, which complicates any interpretation on ladderane fluctuations in this section of the record even further (in addition to our low sampling resolution here). We have therefore included the effect of diagenesis in the first paragraph of section 5 (where we explain why we do not analyse ladderane trends in the older part of the record). These lines are now as follows:

Lines 258-259

'….Unfortunately, the coarse sampling resolution in >160 cal ka BP sediments and low ladderane FA concentrations (due to diagenesis) complicate interpretations of ladderane FA fluctuations in these sediments. Therefore…'

We also agree that anammox may have a lower contribution to total N-loss than denitrification, both in the older section of the record as in the <160 cal ka BP part of the record. We previously tried to elaborate on this in section 5.3, where we argue that $N_2$-production by denitrification and anammox likely occurs in a 70:30 ratio (and hence, denitrification likely contributes more to total N-loss). We realize our intent with this section was not sufficiently clear. Hence we have revised the fourth paragraph of section 5.3 to include the following:

Lines 411-416:

'Moreover, given the average C/N signature of marine OM (106:16; Redfield, 1963), stoichiometric constraints should result in a ratio of $N_2$ production via denitrification and anammox of 71:29 (Koeve and Kähler, 2010). On the one hand, this means that the relative contribution of anammox to $N_2$-production is likely lower than the contribution of denitrification, possibly resulting in a less strong influence of anammox on the $\delta^{15}N$ signal. On the other hand, this means that denitrification and anammox rates should be positively related, in which increased anammox is associated with increased denitrification (Koeve and Kähler, 2010).'

Concerning the suggestion for the figure references. The issue with this is that in results section 4.1 we present the TOC and TN results, while the ladderane FAs are presented in section 4.2. As figure 3 does not include the TOC and TN results (but solely ladderanes) we cannot refer to figure 3 in section 4.1, and thus also not to figure 4. However, we have now resolved this issue by referencing figure 3 in the method section:

Line 218:

'Ladderane concentrations over the entire record (Fig. 3) were used to calculate....'

Consequently, we can now reference Figure in results section 4.1 (line 229):

'.....ranged from 13 to 23 (Fig. 4F–J; S1, Table 3)'.

Additionally, we are now also referencing figure 4 in results section 4.2 (lines 233 and 236).

We hope this sufficiently addresses the raised comments.

*Referee #1*

*The authors of this paper analyzed ladderanes and their degradation products in a sedimentary core collected offshore California and covering the last 500 kyrs. Ladderanes were detected in all the sediment samples and were likely produced within the water column (oxygen minimum zone). The relationship between the abundance of ladderanes and their degradation products (short chain ones) was shown to differ between the Holocene and the*

*rest of the record, suggesting a change in oxygen exposure of these lipids. Overall, this work shows the importance of the anammox process in the California current system and provides a detailed understanding of the N cycling over the last glacial-interglacial periods.*

*This is a very interesting study based on anammox lipid biomarkers, which goes straight to the point and is very well-written. In addition to providing a comprehensive picture of the N dynamics in the zone of study over the last glacial-interglacial periods, this manuscript shows the potential of the ladderanes to reconstruct the anammox process over very long time periods, as these molecules were detected in sediments deposited through the past 500 kyr. I recommend its publication if Biogeosciences after minor revisions, as proposed below*

*Lines 21-22. I would specify that ladderanes and their degradation products were detected.*

**We agree. We have revised it accordingly (lines 21-22): 'Lastly, ladderanes and their SC-products were..'**

*Line 59. Fig. 2 instead of Fig. 1.*

**Thank you for catching this error. We realized, however, that a reference to figure 2 here is not according to Biogeosciences regulations, which state that figures should be referenced in order of their appearance. We have therefore removed the reference to the figure here entirely. Figure 2 is now referenced for the first time (line 103) after figure 1 is referenced.**

*Line 80. Fig. 1 ? Please check the numbering of the figures.* **We have thoroughly checked the numbering of figures and have revised them where necessary.**

*Line 173. I would specify the limit of detection/quantitation of the ladderanes and degradation products in this section.*

**We agree that this should be included. We have included the following sentence in method section 3.5 (lines 187-189):**

**'A detection limit of 30–35 pg injected on-column and a linear response of (r(4) > 0.99) over approximately 3 orders of magnitude was achieved (S1, Table 8a)'.**

*Lines 223-226. Please refer to Fig. 4 here. The colors used to differentiate the individual ladderanes in Fig. 4 are difficult to distinguish. I would also use different symbols.*

**We are now referencing figure 4 int his section (line 229). We have also adjusted the coloring and symbols of figure 4 to enhance the clarity.**

*Lines 235-236. I would also refer to Fig. 4 here.*

**We have adjusted the order in which the ladderane results are presented, so that the ladderane concentrations over the entire record are stated first (Fig. 3) (see lines 233-236). This way, figure 4 can now also referenced in this section (lines 233 and 236).**

*Lines 263-269. The discussion provided here (explanation of the depletion in water column and sedimentary delta15N and presence of ladderanes) should be clarified.*

**We have added further clarifications to this paragraph. More specifically, we have added the following section (lines 271-276):**

**"This means that at ODP site 1012, the sedimentary δ$^{15}$N signal is thought to predominantly derive from the ETNP, and not the ESTNP OMZ. In order to understand the observed ladderane trends in the ODP site 1012 record, it is thus important to establish whether the detected ladderanes reflect a local signal (from the ESTNP OMZ) or whether they are also sourced from the ETNP OMZ core and similarly transported northwards with the CU, towards ODP site 1012. Alternatively, ladderanes could also be synthesized by sedimentary anammox bacteria (Vossenberg et al., 2008)."**

*Line 278. While the transport instead of « And ».*

**This sentence is revised to (line 289) 'Additionally, while the transport…'**

*Lines 280-283. Transport of ladderanes and in situ production of these compounds could both act together ? Why are they considered independently ?*

**Yes that is true, but we clarify our argument in this discussion section, which is based on the observed relative abundances of SC-ladderane (which are similar to those observed in the Arabian Sea where ladderanes are directly deposited on the sea-floor). Due to their labile nature, the transport of ladderanes are likely not the dominant pathway for the occurrence of SC-ladderanes in the record. We agree though that transport of ladderanes cannot be entirely excluded, hence we have added the following to final sentence of this discussion section (line 296):**

**"…..although some contribution of allochthonous or sedimentary anammox cannot be entirely excluded."**

*Lines 293-294. The authors could provide and discuss the C/N ratios and not only TC and TN separately.*

**We agree that a discussion on the C/N ratio would benefit the study. We have therefore included C/N ratios (calculated on an atomic basis) in figure 4 and added the results in section 4.1. In addition, we have included the following sections to the discussion:**

**Section 5.2.1, lines 338-345:**

**'The C/N ratio remains fairly stable throughout MIS 1 to MIS 5c (MN = 16, STD = 2); Fig. 4F), with higher values observed during MIS 6 (MN = 20, STD = 2; discussed in section 5.2.2). Based on stoichiometry, enhanced NO$_3^-$ supply is expected to lower the ratio in phytoplankton biomass (Matsumoto et al., 2020). Yet, changes in nutrient concentrations have been observed to effect the C/P and N/P, but not the C/N ratio (Frigstad et al., 2011). It is therefore not surprising that the increased TN content during mid-MIS 5 is not reflected in the C/N ratio. Also, while the δ$^{13}$C signal (-23 to -22 ‰; Fig. 4I) reflects a typical marine origin of OM, the C/N ratio is higher than commonly observed for marine algae (e.g., Lamb et al., 2006). This is likely caused by preferential remineralization of organic N during the settling of OM from the photic zone (Verardo & Mcintyre, 1994; Schneider et al., 2003).'**

**Section 5.2.2, lines 375-378:**

'MIS 6 and its termination (T2) are further characterized by relatively high C/N ratios (17–23; Fig. 4F). Matsumoto et al., (2020) found, using a global ocean carbon cycle model, that during glacial periods the expansion of sea ice increased global C:N:P ratios. Additionally, taxonomic changes during glacials, in which eukaryotic phytoplankton became more dominant, resulted in $NO_3^-$ depletion (hereby increasing the C/N ratio).'

**Section 5.3, lines 419-421:**

'Moreover, the C/N ratio remains fairly consistent throughout the record (13–19), except during MIS 6 and T2 where it is higher (17–23; Fig. 4F) and variations do not correspond to those observed in ladderane FAs or $\delta^{15}N$.'

*Lines 294-295. This part of the sentence is unclear to me. I would expect an increase in TC and TN with productivity ?* **We have added the word 'elevated' to clarify that it indeed concerns an increase in TOC and TN (line 307).**

*Line 296. Please remove « interestingly » (no opinion on your results). « Are not the highest ».* **We have removed the word interestingly and have adjusted it to:**

**Line (309): 'In contrast to ladderane FAs, concentrations of…'**

*Line 297. Please specify the number of the figure.* **Thank you for spotting this error, we have included the number of the figure (i.e. Fig. 3; line 310).**

*Line 304. « This suggests that » rather than « This appears to ».* **We have removed 'this appears to' and changed it to 'this indicates that' (line 318).**

*Lines 307-308. What is the link between the two paragraphs ?*

**The paragraph starting at line 308 explains the possible mechanism (i.e. an intensified OMZ) for the inferred reduced oxygen exposure discussed in the preceding paragraph. We have revised the first sentence of the paragraph to better underline the relationship with the preceding paragraph, as follows (line 321):**

**'Reduced oxygen exposure is likely to have resulted from an intensified OMZ; Lembke-Jene et al. (2018) showed…'**

*Lines 308-312. This could also explain the low exposition of ladderane FAs to oxygen over this period ?*

**Yes, we agree. The purpose of this paragraph is to provide an explanation for the reduced oxygen exposure of ladderane FAs during the Holocene, but we realize this was not clear. We hope that with the revision presented in response to the previous comment, the relationship between this and previous paragraph is more clear (i.e. how an intensified OMZ led to more reduced oxygen exposure of ladderane FAs during the Holocene).**

*Lines 313-314. The concentrations peak just before MIS 5ᵉ ?*

**Both, as we see it. Indeed, there is a peak at the end of MIS 6 (right before MIS 5e), which is discussed in section 5.2.2 (which deals with the two most recent glacial periods). But there is also a peak during MIS 5e.**

*Lines 331-333. I would show the relationship between the concentrations in ladderanes and TC and TN to support the hypothesis of the co-variation with paleoproductivity.*

**In this sentence we were solely referring to the co-variation between those proxies during MIS 5, and not over the entire record. We realize this was not sufficiently clear. Estimating whether a (linear) relationship exists between ladderanes and productivity proxies during MIS 5 is unfortunately not possible, as the sample size would be too small (<20). We realize the word 'co-variation' seems to indicate a significant relationship. Hence, we have adjusted this sentence to clarify our meaning, as follows:**

**Line 349: 'As such, the co-occurrence of ladderane FA and paleo-productivity proxies maxima during MIS 5…'**

**Also, to further clarify: over the entire record, there is no significant relationship between ladderanes and productivity proxies. Discrepancies between their concentrations are for example seen during MIS 2 and MIS 6. These discrepancies during these glacial periods are addressed in more detail in section 5.2.2. with the following:**

**Lines 378-380: 'At the same time, decreased upwelling during glacial periods in the North Pacific (Worne et al., 2019) may have also lowered nutrient availability. Low nitrogen availability is reflected in relatively low TN concentrations in this record (Fig. 4H).'**

*Lines 333-335. This part is unclear to me and should be clarified.*

**We will move this section to the next paragraph and revise it to the following:**

**Lines 351-354: 'Remineralization of increased phytoplankton biomass may consequently also have led to more reduced local conditions, favouring anammox. This local signal would not have been recorded in the western part of the North Pacific, where intermediate waters were oxic (Matul et al., 2016).'**

*Lines 335-336. Please further explain what are these discrepancies exactly.*

**We have revised this sentence as follows (lines 354-355): 'The relatively subdued $\delta^{15}$N signal during mid-MIS 5, and consequent implications for our understanding of the N-cycle in the CCS are further discussed in section 5.3.'**

*Lines 373-374. This correlation is not valid over all the record (this is a general trend).*

**We have adjusted this line to more precisely describe the results by the Liu et al. (2005) study as follows (line 396): 'The cross-correlation for both $\delta^{15}$N–$\delta^{18}$O and $\delta^{18}$O–SST at ODP site 1012 (Liu et al., 2005) indicates that….'.**

*Lines 380-381. These discrepancies are seen more during MIS 5 than MIS 3.*

**We agree with this. Also, since we solely present the discrepancy with MIS 5 in section 5.2.1, we have removed MIS 3 from this sentence (line 404).**

*Lines 382-387. Please explain more clearly how it can be detected that anammox and denitrification are out-of-phase.*

**We apologize for this unclarity. We did not mean to state that they are out-of-phase. Rather, we meant to indicate that this may be an option, but that the known literature indicates this is actually not a reasonable explanation for the discrepancies between the ladderane and $\delta^{15}$N record. Also, in response to the next comment, and in response to comments of both reviewers and the Associate editor, we have revised the structure of this discussion section and added some clarifications, to better highlight our conclusion. We made the following revisions (lines 405-423):**

**'This suggests that increased anammox does not always correspond to increased N-loss, possibly via simultaneously reduced denitrification rates (Koeve and Kähler, 2010). Yet, Babbin et al., (2014) showed, using incubations from the ETNP OMZ, that both denitrification and anammox are limited by OM supply, and their rates increase in response to the addition of OM. Moreover, these authors showed that both denitrifiers and anammox bacteria are similarly inhibited by oxygen in the marine environment, at DO concentration around 3 to 8 μmol L-1 (Babbin et al., 2014). As such, both anammox and denitrification should respond similarly to changes in DO and OM in the CCS.**

**Moreover, given the average C/N signature of marine OM (106:16; Redfield, 1963), stoichiometric constraints should result in a ratio of N2 production via denitrification and anammox of 71:29 (Koeve and Kähler, 2010). On the one hand, this means that the relative contribution of anammox to N2-production is likely lower than the contribution of denitrification, possibly resulting in a less strong influence of anammox on the $\delta^{15}$N signal. On the other hand, this means that denitrification and anammox rates should be positively related, in which increased anammox is associated with increased denitrification (Koeve and Kähler, 2010). Potentially, anammox and denitrification could be unsynchronized (as indicated by differences between the ladderane and $\delta^{15}$N records) in response to variations in the C/N ratio of OM. Localized variations in the C/N signature may result in different relative contributions. Yet, integrating these variations over space and time should obtain a similar ratio (Dalsgaard et al., 2012; Ward, 2013; Babbin et al., 2014). Additionally, the C/N ratio remains fairly consistent throughout the record (13–19), except during MIS 6 where it is higher (17–23; Fig. 4F) and variations do not correspond to those observed in ladderane FAs or $\delta^{15}$N. As such, given the temporal resolution of the record (which does not cover seasonality), denitrification and anammox intensities are expected to fluctuate in-tandem.**

**Consequently, variability in $\delta^{15}$N of the CCS sedimentary record may, at times, simply not relate directly to changes in denitrification and/or anammox rates. Reconstructions of…'**

*Lines 388-393. I don't see the aim of this paragraph and where the authors want to go.*

With this paragraph, we aim to exclude DO concentrations and OM supply as possible reasons for the observed differences between the $\delta^{15}$N and ladderane FA record. To clarify the purpose of this paragraph, we have revised this discussion section, as described in the previous comment.

*Line 393. « can also be supplied »*

We have removes this sentence (line 409), as it distracts from the main message of the paragraph and was repetitive (a similar sentence is also present in discussion section 5.2.1, line 349).

*Line 403. Please provide the determination coefficient of this correlation.*

We apologize, we have wrote this down incorrectly. The Liu et al. (2005) study does not show a direct correlation between $\delta^{15}$N and U$^{K'}_{37}$ -based SST but rather a cross correlation for both $\delta^{15}$N–$\delta^{18}$O and $\delta^{18}$O–SST, which is now also revised in line 396. We have adjusted this sentence to now reflect a result from Liu et al. (2008), rather than Liu et al. (2005). This line is now as follows (433-434):

'Given the phase-relationship between the $\delta^{15}$N and U$^{K'}_{37}$ -based SST records of the CCS (Liu et al., 2008) and ….'

*Line 404. « it may be reasonable »* This is now adjusted (line 434).

**Referee #2**

*This study investigates the presence of anammox bacteria in the OMZ of the Eastern Subtropical North Pacific by means of ladderane lipid analysis. In total, 69 sediment samples were analysed from ODP site 1012 covering the past ca 500 kyrs. Short-chain ladderanes are interpreted as measure of oxygen exposure and thus changes in OMZ strength. Ladderane lipids were detected in all samples, however, the older parts of the sediment cores showed very low concentrations and almost no variation, which makes any profound interpretation on past N-cycling difficult. The focus of the current study is therefore restricted to the past ca 160 kyrs. As far as I know this is one of the very few studies using ladderane lipids as proxy for past anammox activity and indicating an alternative pathway of N-loss besides denitrification. It also reports the oldest evidence for anammox in the sedimentary record and I recommend publication after minor revision.*

*Title: Ladderane abundances in the older parts of the core are extremely low and basically show no variation. Please consider changing the title to '...late Quaternary' or '..during the last 160 kyrs'.*

We agree that changing the title to '….late Quaternary' better describes the time-span that is discussed in the manuscript and have changed the title accordingly.

*Line31-32: maybe delete sentence as you repeat in Hydrographic setting Line 118-119.*

We agree that this is repetitive. However, as we introduce various studies within the introduction that either investigated the ESTNP or the ETNP, we feel it is necessary to include this sentence in the introduction. Nonetheless, we will remove the coordinates

within the sentence, which are now only mentioned in the hydrographic section (to limit some of the repetitiveness).

*Line 71: you may want to add the study by Brunner et al. (2012, PNAS 110(47), 18994-18999), who first showed N isotope effects in anammox bacteria (K. stuttgartiensis).*

**We agree it would be appropriate to reference this study. We have adapted the sentence as follows (lines 70-71):**

**'Enrichment cultures of anammox bacteria have, however, shown that they induce a similar isotope fractionation effect (Brunner et al., 2013), with that of *Ca.* Scalindua spp. being +16 to +30 ‰ (Kobayashi et al., 2019).'**

*Line 77: Ganeshram et al. (2000) is missing in the reference list.*

**Thank you for pointing this out. The Ganeshram paper has been added to the reference list.**

*Line 101ff: what are modern SST and subsurface temperatures (OMZ)? I think it's worth adding this information as you discuss temperatures later.*

**While we agree that this information would provide a greater understanding of the system, it would require quite a lengthy explanation as decadal-, annual- and seasonal-scale natural oscillations of the climate system lead to a large variation in temperatures within the CCS and the ENP OMZ. As our study does not aim to compare reconstructed NL$_5$-temperatures with modern-day temperatures, we decided not to include an introduction on modern-day temperatures in the Hydrographic section (section 2). Even so, we do mention modern-day temperatures in the context of the discussion in section 5.1, but solely to establish whether the NL$_5$-derived temperatures relate better to water column or sea-floor temperatures. We feel this is sufficient for the purpose of the manuscript.**

*Line 112: CU already introduced in line 59.*

**The sentence will be altered to only include the abbreviation. We will make the same adjustment for line 105 in the Hydrographic section.**

*Line 221ff: results are generally reported in past tense.*

**We have adjusted the results section so that results are now reported in the past tense.**

*Line 224: delete 'The content of'*

**This is deleted (line 227).**

*Line 235: maybe add here the information that ladderane concentrations without the internal standards are significantly lower than those calculated with internal standard.*

**We have added the following information in results section 4.2.1 (lines 236-240):**

'Concentrations calculated without the use of the internal standard (Hopmans et al., 2006; see section 2.5) are reported in S1 (Table 8b) and were a factor 1.2 and 1.3 lower for [3]-(SC-)ladderanes [5]-(SC-)ladderanes, respectively. Concentrations calculated with the two quantification methods showed a strong positive linear relationship of $R^2$ = 0.88 and 0.89 for [3]-(SC-)ladderanes and [5]-(SC-)ladderanes, respectively (Fig. S2.2).'

*Line 235-236: maybe add a mean NL5 value and also mention that highest values are recorded in the older core section. I wonder why? Maybe higher uncertainty because of low concentrations?*

We have added the following section to this sentence (line 241):

'…with highest values observed in >160 cal ka BP sediments (S1, Table 6).'

We agree that it is certainly possible that these higher values are the cause of a higher uncertainty associated with the lower concentrations. While we have thought about including a discussion on this possibility in the manuscript, we have opted not to do so because we solely use $NL_5$-derived temperatures in order to elucidate whether the ladderane signal derives from water column or sedimentary anammox (see section 5.1) and therefore a discussion on factors influencing the $NL_5$-index in >160 cal ka BP sediments exceeds the purpose of the manuscript. We hope a more in-depth discussion on this can be included in future work by either ourselves or others.

*Line 243-244: maybe replace by '..ladderane FAs and their degradation products..'*

We will revise this sentence as follows (lines 248-249): '…$C_{18}$[3]-, $C_{18}$[5]-, $C_{20}$[3]- and $C_{20}$[5]- ladderane Fas and their short chain $C_{14}$[3]-, $C_{14}$[5]-, $C_{16}$[5]-products were…'

*Line 259/Fig. 4: I suggest to show NL5 values instead of temperatures, which seem a bit random. Add a threshold value according to Rattray et al. (2010) that indicates water column vs. sedimentary anammox.*

We agree that it is better to show the original NL5-index values. We have therefore adjusted the figure accordingly. As for adding a threshold value: to the best of our knowledge, Rattray et al., (2010) does not provide an exact threshold value for sedimentary anammox. We have therefore included the following sentence and hope this sufficiently addresses the comment (lines 285-287):

'According to the $NL_5$-calibration by Rattray et al., (2010), $NL_5$ indices within this range more closely reflect water column rather than sedimentary anammox bacteria'.

*Line 248ff: the authors argue that this lack of variation may be due to the low sampling resolution. Alternatively, a weaker OMZ or predominance of denitrification in the OMZ may also explain the general low abundance of anammox bacteria and thus ladderane lipids. The d15N record of Liu et al. (2015) still shows variation between ca 160-500 kyr so maybe anammox was just minor? Or due to strong degradation of labile ladderanes? This was also indicated in the Arabian Sea anammox record.*

We apologize for the unclarity. We did not aim to argue that there is a lack of variation in ladderane concentrations in >160 cal ka BP sediment, but that the resolution of our

record in this time-frame is not sufficient to assess whether any variability exists. In addition, we agree that a strong degradation of ladderanes also complicates interpretation of any variability. We have therefore adjusted this section as follows (lines 253-254):

'Unfortunately, the coarse sampling resolution in >160 cal ka BP sediments and low ladderane FA concentrations (due to diagenesis) complicate interpretations of the variations in ladderane FAs. Therefore, analysis of trends in ladderane concentrations over (inter)glacial cycling is limited to <160 cal ka BP sediments.'

We have also further elaborated on the idea that anammox likely imposes a less strong influence on the d15N record than denitrification by including the following lines in section 5.3:

Lines 411-416:

'Moreover, given the average C/N signature of marine OM (106:16; Redfield, 1963), stoichiometric constraints should result in a ratio of $N_2$ production via denitrification and anammox of 71:29 (Koeve and Kähler, 2010). On the one hand, this means that the relative contribution of anammox to $N_2$-production is likely lower than the contribution of denitrification, possibly resulting in a less strong influence of anammox on the $\delta^{15}N$ signal. On the other hand, this means that denitrification and anammox rates should be positively related, in which increased anammox is associated with increased denitrification (Koeve and Kähler, 2010).'

*Line 256/Fig. 5: add a, b, c,... to the panels.*

This is now added to the panels.

*Line 297: where is Fig. X?*

Thank you for spotting this error. We have adjusted the reference of this figure (which is figure 5A).

*Line 308-312: this paragraph seems a bit out of place. It relates to the interpretation before (lines 296ff)? Then make the link.*

This paragraph explains the possible mechanism via which ladderanes could have experienced a decreased oxygen exposure. We agree that the transition to this paragraph was unclear. We have therefore included the following phrasing, and hope this better explains why this paragraph follows the preceding one (line 321):

'Reduced oxygen exposure is likely to have resulted from an intensified OMZ; Lembke-Jene et al. (2018) showed…'

*Line 313ff: there are distinct maxima in SST, d18O and d15N during MIS 5e (Fig. 4) but not in ladderanes. There is some variation during MIS 5 but there is no clear trend as seen in d15N. d15N shows a clear pattern of higher values during warm intervals 5a, c, e and lower values during cold intervals 5b, d, reflecting changes in denitrification. This is not seen in*

*ladderanes, maybe due to the low sampling resolution (for instance there is only one datapoint covering MIS 5d).*

**We address the discrepancies between these proxies in detail in section 5.3. We hope this sufficiently addresses the raised points.**

*Line 318: You mean 'during MIS 5b and d, intermediate waters... were oxic'*

**No not exactly. The Matul et al. (2016) reference mentioned in this sentence revealed that intermediate waters in the western North Pacific were oxic throughout MIS 5b-d (so including MIS 5c). We have revised this paragraph to better explain how we can still observed high ladderane concentrations at this time, while taking into account oxic western North Pacific intermediate waters (lines 352-354):**

**'Remineralization of increased phytoplankton biomass may consequently also have led to more reduced local conditions, which would also favour anammox. This local signal would not have been recorded in the western part of the North Pacific, where intermediate waters were oxic (Matul et al., 2016).'**

*Line 351ff: both deglaciations are only represented by a single data point. Particularly during the last deglaciation, a more gradual increase in ladderanes toward the Holocene following d15N is also likely.*

**We agree that the conclusions made in this paragraph are probably not sufficiently supported by the data, as indeed we only have one data point for each deglaciation. We have therefore decided to remove this paragraph, and now do not make any inferences on anammox trends during deglaciations. Instead, we have added the following paragraph, which we believe fits the discussion of this section better (lines 383-388):**

**'While enhanced anammox in response to deoxygenation during glacial maxima is at odds with previous assessments of N-loss in the CCS (e.g., Liu et al., 2005), deoxygenation of the Pacific is consistent with recent paleo-proxy studies (Anderson et al., 2019; Lu et al., 2016) and modelling results (Matsumoto et al., 2020). According to these studies, many parts of the glacial ocean, including the equatorial Pacific, had substantially lower DO during the last glacial period than today. This fits with increased ladderane FAs at this time, which suggests N-loss in the CCS was likely more intense during glacial maxima than previously assumed.'**

*Line 370: I guess that one of the potential reasons for higher ladderane concentrations compared to the Arabian Sea record may relate to analytical improvements, i.e. use of internal standard.*

**We agree that this may indeed contribute to the higher concentrations observed in our record, and that stating that 'ladderane concentrations are higher than in the Arabian Sea' without giving any context on why this is, should be prevented. We have therefore adjusted this sentence to the following (lines 393-395):**

**'However, the occurrence of ladderane FAs throughout our CCS record now shows that anammox was (also) responsible for N-loss and thus contributed, at least partially, to the sedimentary $\delta^{15}N$ record.'**

*Line 378ff: what do you think are the main differences to the anammox record from the Arabian Sea OMZ, where ladderanes closely follow TOC and d15N on the glacial-interglacial cycle? A generally weaker/thinner OMZ and higher dynamics of N-cycling processes in the ESTNP? Incomplete utilization of nitrate during periods of enhanced upwelling?*

**Yes, indeed we think that the $\delta^{15}N$ record of the CCS does not solely reflect changes in N-loss rates, but also $NO_3^-$ availability and $NO_3^-$ assimilation by phytoplankton. An incomplete utilization of nitrate during periods of enhanced upwelling leads to a subdued $\delta^{15}N$ signal, hereby leading to differences in variability between the ladderane and $\delta^{15}N$ records. We outline our arguments for this idea in section 5.3, which we have revised sightly to better highlight our arguments as follows (lines 423-435):**

**'Consequently, variability in $\delta^{15}N$ of the CCS sedimentary record may, at times, simply not relate directly to changes in denitrification and/or anammox rates. Reconstructions of N-loss using sedimentary $\delta^{15}N$ depend on the assumption that there was complete biological utilization of $NO_3^-$ by phytoplankton. However, during periods of high upwelling intensity (as likely occurred during mid-MIS 5; see section 5.2.1), the high $NO_3^-$ availability may result in incomplete $NO_3^-$ assimilation. This allows for the preferential uptake of $^{14}N$ by primary producers, resulting in a pool of $\delta^{15}N$ depleted OM available for heterotrophic denitrification (Tesdal et al., 2013). Hence, at times of high $NO_3^-$ supply, incomplete nitrate assimilation would have quenched the $\delta^{15}N$ signal, even if denitrification was as intense as during periods of low $NO_3^-$ availability. Moreover, a study by Altabet and Francois (1994) showed that sedimentary $\delta^{15}N$ in the equatorial Pacific records the isotopic enrichment of near-surface $NO_3^-$ via depletion by phytoplankton, in which enriched $\delta^{15}N$ values are associated with reduced $NO_3$ availability for phytoplankton assimilation. Also, in the South Pacific, $NO_3^-$ concentrations have been found to affect the $U^{K'}_{37}$ index (Placencia et al., 2010). Given the phase-relationship between the $\delta^{15}N$ and $U^{K'}_{37}$ -based SST records of the CCS (Liu et al., 2008) and the discrepancies between the $\delta^{15}N$ and ladderane records, it may be reasonable to conclude that the CCS sedimentary $\delta^{15}N$ fluctuations (also) record variations in $NO_3^-$ assimilation by phytoplankton.'**

*Line 382: maybe add C/N ratios to Fig. 4.*

**Referee #1 also made this suggestion. We agree that an inclusion of the C/N ratio would benefit the study. We have therefore included C/N ratios (calculated using their atomic mass) in figure 4, added the results in section 4.1 (line 226) and now discuss the ratio in various sections (see response to comment of Referee #1).**

*Line 387: if denitrification and anammox are expected to fluctuate in tandem, you would expect similar patterns in both d15N and ladderanes, right? This is not the case. Maybe anammox contributes to N-loss only to a minor extent?*

**In section 5.3, we argue why $\delta^{15}N$ and ladderanes do not always show the same pattern, even though anammox and denitrification are inferred to fluctuate in-tandem. The main reason is that the $\delta^{15}N$ record is shaped by many processes. We believe that the $\delta^{15}N$ record of the CCS does not solely reflect an N-loss (or denitrification) signal, but is amongst others also shaped by incomplete nitrate utilization during periods of intense**

**upwelling. These arguments are now outlined more precisely in section 5.3 (see also the response to previous comments).**

*Line 431: I understand that ladderane analysis is very time-consuming but this is not a high-resolution record compared to Liu et al. (2005) and Herbert et al. (2001). Please change*

**Agreed. We have adjusted the sentence as follows (line 461): 'The CCS record shows a continuous presence of ladderane FAs over the last two interglacial-glacial transitions...'.**